# Luminescent Molecularly Imprinted Polymers Based on Covalent Organic Frameworks and Quantum Dots with Strong Optical Response to Quinoxaline-2-Carboxylicacid

**DOI:** 10.3390/polym11040708

**Published:** 2019-04-17

**Authors:** Ying Zhang, Dianwei Zhang, Huilin Liu

**Affiliations:** Beijing Advanced Innovation Center for Food Nutrition and Human Health, Beijing Engineering and Technology Research Center of Food Additives, Beijing Technology and Business University, 11 Fucheng Road, Beijing 100048, China; zhangying940907@163.com (Y.Z.); zhangdianwei1205@163.com (D.Z.)

**Keywords:** molecularly imprinted polymers, covalent organic frameworks, quantum dots, quinoxaline-2-carboxylicacid, luminescent

## Abstract

Three-dimensional molecularly imprinted polymers (MIPs) based on quantum dots-grafted covalent organic frameworks (QDs-grafted COFs) are reported in this study. The compound 1,3,5-triformylphloroglucinol-P-phenylenediamine was used as COF material to react with the amino-modified CdSe/ZnS QDs by Schiff-base reactions. The amino-derived QDs reacted with quinoxaline-2-carboxylicacid (QCA) via a non-covalent interaction. The system combines the advantages of MIPs, COFs, and QDs for highly sensitive and selective QCA detection. The MIPs based on QDs-grafted COFs showed good chemical selectivity and thermal stability, as well as consistency in QCA optosensing. Under optimal conditions, the detection limit for QCA in meat and feed samples was 0.85 μmol L^−1^, over a linear concentration range of 1–50 μmol L^−1^. The current findings suggest a potential application of MIPs based on QDs-grafted COFs for the detection of trace levels of hazardous chemicals for food safety and environmental control.

## 1. Introduction

Luminescent covalent organic frameworks (COFs) with the porosity of COFs and intrinsic optical properties, which can transform the chemical signal produced by the host–guest interaction into detectable changes in luminescence, have attracted wide attention. Although various topological diagrams of building blocks with predesigned geometry and symmetry by virtue of dynamic covalent bonds have been developed for COFs synthesis [1,2,3,4,5], luminescent COFs remain a challenge. Most layered, two-dimensional COFs (2D-COFs) with uniform and perforated pores exhibit conjugated 2D layers and periodic columnar π–π arrays, which trigger the thermal decay of photoexcited states and result in less- or non-emissive luminescence [6]. The π–π accumulation-caused quenching mechanism has been explained by the quick dissipation of the COFs excitation energy [7,8]. Crystalline multiple-component COFs are constructed by both a heterogeneous and a homogeneous distribution of π-conjugated vertex units of dehydrobenzoannulene and exhibit unique solid-state optical properties [9]. Dalapati et al. have prepared a highly emissive COFs using an aggregation-induced emission-active chromophore to the π-frameworks. It uses an aggregation-induced emission mechanism to interact with the intralayer covalent bonding, and the interlayer non-covalent π-interactions work to reduce rotation-induced thermal decay of the photo-excited state [10]. The highly luminescent COFs are first achieved, which use a high sensitivity sensor to reduce the detection limit of ammonia at the sub-ppm level. This framework laid the foundation for COFs application in the luminescent sensor field. The highly luminescent COFs sensors rely on optical detection that confers them high quantum yields, selectivity and sensitivity, and short response time and facilitates data collection. Guest luminescent materials with good optical properties are introduced into COFs maintaining host COFs properties and their original structures and providing an interesting alternative way to get ideal highly luminescent COFs.

Semiconductor quantum dots (QDs) are good inorganic chromophores with efficient size-dependent broadband absorption, high extinction coefficients, and intense size-tunable narrowband fluorescence. They have been widely used in bioanalysis [11], particularly because of their brightness and high stability with respect to photobleaching [1,3,12,13]. Optosensing via charge/energy transfer interactions with analytes have allowed QDs to become sensitive response elements. When modified with organic ligands of three octyl phosphine oxide, QDs have high fluorescent quantum yields and stability as opto-sensors. However, they cannot identify analytes accurately if specific recognition groups are not attached to the QD surface.

In order to improve the selectivity and target recognition, molecular imprinting technology is used in the study. The resulting molecular imprinted polymers (MIPs) have gained wide attention and attained significant applications in selective enrichment and extraction of targets from complex matrices [14,15,16]. MIP preparation used functional monomers and crosslinkers in the presence of a unique template molecule, and then the binding cavities were generated on the basis of shape matching and functional interactions following the removal of a template molecule [17]. A reverse microemulsion method for MIP preparation exhibited specificity for the template molecule and improved the accuracy. A three-dimensional network structure was formed on the QD surface by hydrolysis of silylating reagents, which provides surface binding sites for recognition groups, inhibits QD photo-oxidation, improves fluorescence stability, and prevents diffusion from their toxic heavy metal ions.

Quinoxaline-2-carboxylicacid (QCA) was chosen as the target to illustrate the use of the protocol. It is a metabolite of carbadox in tissues [18] and is used as a feed-additive for aquaculture and livestock. Because of possible carcinogenic and mutagenic effects, several QCA analytical methods have been developed. Duan et al. used QCA-imprinted solid-phase extraction (SPE) coupled with high-performance liquid chromatography (HPLC) for sensitive and selective detection of QCA residues [19]. However, HPLC is costly and involves time-consuming processing and tedious pretreatment steps that require large amounts of organic reagents and skilled operators. Therefore, a more convenient and rapid sensor technology is of great interest. Shah et al. studied the QCA derivative 6-(1-hydroxynaphthalen-2-yl) benzo[f] QCA, using an electrochemical technique on the surface of a glassy carbon electrode over a wide pH range [20]. Yang et al. also used an electrochemical determination of QCA, based on a functional double layer of poly(pyrrole) composite via electro-polymerization and molecularly imprinted poly(o-phenylenediamine) [21]. Electrochemical sensing can avoid the disadvantages of HPLC, but long-term fabrication processes and poor adhesion on transducer surfaces can limit detection accuracy and reproducibility.

In the study, we report a reverse microemulsion strategy for the synthesis of MIPs based on QDs-based COFs for optosensing QCA with high sensitivity and selectivity. QDs were used as an optical response element, and COFs were used as an optosensing platform to combine the MIP reverse microemulsion strategy and obtain high selectivity. The sensitivity was derived from the dual-signal amplification by high QD fluorescence quantum yields and the large COF-specific surface area. In addition, the prepared QDs-grafted COFs exhibited good repeatability and high thermal and chemical stabilities. Fabrication of MIPs based on QDs-based COFs via reverse microemulsion is also promising in its application as a platform for QCA detection.

## 2. Materials and Methods

### 2.1. Materials

QCA (95%) was purchased from Adamas Reagent Co., Ltd., Shanghai, China. Olaquindox (QLA; 99%), and carbadox (CBX; 98.5%) was purchased from Dr. Ehrenstorfer GmbH Co., Ltd., Augsburg, Germany. Mequindox (MEQ; 99.6%) was purchased from the Institute of Veterinary Drug Control, Beijing, China; 3-Methylquinoline-2-carboxylic acid (MQCA; 95%) was purchased from Aladdin Bio-Chem Technology Co., Ltd., Shanghai, China. CdSe/ZnS QDs with 460 nm excitation and 605 nm fluorescence emission were purchased from Jiayuan (Wuhan, China); 1,3,5-triformylphloroglucinol (TP; 99%) was purchased from Strem Chemicals, Inc., Newburyport, MA, USA. P-phenylenediamine (Pa) and mesitylene (99%) were purchased from Shanghai Macklin Biochemical Co., Ltd., Shanghai, China; 3-Aminopropyl triethoxysilane (APTES; 98%), acetic acid (99.5%), Triton X-100 (99.5%), and cyclohexane (99.5%) were purchased from Sinopharm Chemical Reagent Co., Ltd., Beijing, China. Dioxane (99.5%) and tetraethyl orthosilicate (TEOS; 98%) were purchased from J&K Scientific Ltd., Beijing, China. Doubly de-ionized water (DDW; 18.2 MΩ cm^−1^) was obtained from a Water Pro water purification system (Labconco, Kansas City, MO, USA).

### 2.2. Instrumentation

Fluorescence measurements were performed with an FL-4500 spectrofluorometer (Hitachi, Tokyo, Japan). The surface morphologies were imaged by scanning electron microscopy (SEM, SU1510, HITACHI, Tokyo, Japan). Fourier-transform infrared (FTIR) spectra (4000–400 cm^−1^) were obtained using a Bruker Vertex 70vxrd spectrophotometer (Bruker, Vertex 70vxrd, Germany). Ultraviolet–visible (UV–vis) absorption spectra (200–800 nm) were recorded on a Cary Series UV spectrometer (Agilent, Santa Clara, CA, America). Thermogravimetric analysis (TGA) was performed in air with a PTC-10A analyzer (Rigaku, Tokyo, Japan) from room temperature to 700 °C at a ramp rate of 26 °C min^−1^. X-ray diffraction (XRD) patterns were collected using a Rigaku D/max-2500 diffractometer (Rigaku, Tokyo, Japan).

### 2.3. Synthesis of TpPa COFs

The typical Schiff-base reaction was used for the synthesis of TpPa COFs [22]. In a 500 mL capacity conical flask, 0.3 mmol of 1,3,5-triformylphloroglucinol (Tp) and 0.45 mmol of P-phenylenediamine (Pa) were mixed, then 1.5 mmol of acetic acid and 3 mL of mesitylene/dioxane (1:1) were added successively. Then, the mixture was sonicated for 10 min, after keeping it at 120 °C for three days with nitrogen sealing. After centrifugation, a red-colored precipitate was obtained, and anhydrous acetone was used to wash the precipitate six times. Finally, it was dried under a vacuum at 180 °C for 24 h. 

### 2.4. Synthesis of MIPs Based on QDs-Grafted COFs 

The MIPs based on QDs-grafted COFs were prepared by reverse microemulsion [23]. Triton X-100 (180.0 mL) and cyclohexane (750.0 mL) were added into a 25 mL round-bottom flask, and then stirred for 15 min at room temperature. CdSe/ZnS QDs (1.0 mL), silane reagent of TEOS (50.0 μL), and aqueous ammonia (25 wt%, 100.0 μL) were then added to the flask. After stirring for 2 h, APTES (20.0 μL) and TpPa (0.5 mg) were added to the mixture, which was stirred for another 2 h. QCA (3.0 mg) was dissolved in ethanol solution (10.0 mL) and added (100.0 μL) to the flask. The mixture was sealed and stirred for 24 h. After pre-polymerization, the product was purified by centrifugation and then washed twice with acetone and DDW. In all of the washing procedures, the mixture was centrifuged to remove the supernatant at 5000× *g* for 10 min, and then the precipitate was re-dispersed in the next solvent. After purification, the products were washed with methanol until no template molecules were found using UV–vis spectrophotometry. Lastly, the products were dried under vacuum at 40 °C for 10 h. The products without QCA were prepared by the same procedure.

### 2.5. Fluorescence Measurement

All of the measurements of fluorescence intensity were performed under the same conditions. The excitation and emission wavelength were set at 460 nm and 610 nm, respectively. The MIPs based on QDs-grafted COFs were mixed evenly for FL-4500 spectrofluorometer analysis.

During the detection, 3 mL of target standard solution at a specific concentration was added into a quartz cell before 1.0 mg of MIPs based on QDs-grafted TpPa COFs was added. After mixing, fluorescence intensity detection was performed using the FL-4500 spectrofluorometer. 

### 2.6. Meat and Feed Samples

Recently, QCA was reported at trace levels in beef, pork, fish, pork feed, cattle feed, and fish feed samples. Therefore, meat and feed samples were collected from a local supermarket. We homogenized the samples by a high-speed food blender, and packed them (5.00 ± 0.01 g) into a 100 mL centrifuge tube. QCA was added at three concentrations: 0, 5, 10, and 20 μmol L^−1^. Next, 5 mL of a solution containing 5% metaphosphoric acid (*w*/*v*) and 20% acetonitrile (*v*/*v*) was added. After the above mixture was mixed for 5 min, 10 mL of acetonitrile was added. Then, it was centrifuged at 5000× *g* for 10 min, and the supernatant was collected and evaporated to near dryness at 40 °C. Finally, the residue was re-dissolved in 1 mL of methanol and diluted to 100 mL with phosphate-buffered saline (pH 7.5).

## 3. Results and discussion

### 3.1. Synthesis of MIPs Based on QDs-Grafted COFs for Detection of QCA

In the present study, novel MIPs based on QDs-grafted COFs were prepared for a sensitive and selective detection of QCA. The structure and synthesis procedures of the QDs-grafted COFs are depicted in Figure 1. The preparation of TpPa COFs was carried out by the Schiff-base reactions of Tp with Pa with the formation of a crystalline framework, in the presence of acetic acid using 1:1 mesitylene/dioxane as the solvent. The chemical stability of the COFs was obtained by an irreversible enol-to-keto tautomerization. A crystalline framework was formed that enhanced chemical stability. During the preparation, a one-pot reverse microemulsion strategy was used. With the surfactant interactions of reverse microemulsion, the QDs were encapsulated in aqueous domains by magnetic stirring. With the further hydrolysis and condensation reactions of the silane reagent of APTES and TEOS, silica nanospheres were fabricated. APTES was used as the functional monomer to react with the target QCA using H-bonding reactions. The APTES-modified QDs complex was further reacted with the TpPa COFs using the Schiff-base reaction. The cavity sites were left after the QCA was extracted from the QDs-grafted COFs using methanol to disrupt the H-bonding. Prior to the removal of the QCA, the fluorescence intensity of the QDs-grafted COFs was 37.4%; after removal, it was 99.2% (Appendix A). The repeatability of QDs-grafted COFs was investigated for a QCA concentration of 1 μmol L^−1^ over five measurements. Relative standard deviations (RSD) of 1.81%, 1.36%, 2.15%, 1.42%, and 2.23% were obtained by spectrofluorometry, indicating good optosensing repeatability.

In addition, the MIPs based on QDs-grafted COFs had high fluorescence quantum yields due to the hydrophobic QDs and the polymerization method of reverse microemulsion. The quantum yield was calculated by using Rhodamine B as a reference and was 33.5%.

### 3.2. Characterization of MIPs Based on QDs-Grafted COFs

The TEM images in Figure 2a showed that the QDs were uniformly distributed and sized. QDS size was about 3–5 nm in diameter. In Figure 2b, the SEM images of TpPa COFs revealed a tree-like morphology. We also used SEM to observe MIPs based on QDs-grafted COFs (Figure 2c), which revealed a rough morphology of highly uniform spheres. It indicated that when QCA was removed from the surface of the QDs-grafted COFs, it left recognition sites. The tree-like morphology has been seen in QDs-grafted COFs. A lot of CdSe/ZnS QDs were aggregated in the leaves of the COFs.

The FTIR spectra of COFs (Appendix A) and MIPs based on QDs-grafted COFs (Appendix A) was used to judge the occurrence of the polymerization reactions. It can be seen from Appendix A that the peaks at 1690 cm^−1^ and 1250 cm^−1^ were the stretching vibration of C=O and C–N, respectively, for the TpPa, and a strong peak at 1577 cm^−1^ was attributed to the C=C stretch in the keto form. Characteristic peaks at 1616 cm^−1^ were attributed to the decreased C=O stretching frequencies due to the covalent interactions of TpPa COFs and QDs. Therefore, the QDs were attached to the COF surface successfully.

An XRD pattern from MIPs based on QDs-grafted COFs is shown in Appendix A. It revealed a cubic structure of the peaks at 26.46°, 44.10°, and 51.91°, which were similar to QD patterns. The XRD pattern of QDs was reported in previous work [24]. This confirmed that QDs were grafted on the COFs.

The TGA analysis of MIPs based on QDs-grafted COFs is shown in Appendix A. The TGA data exhibited only 3.57% weight loss under 100 °C, while the loss was 11.75% over the range 100–300 °C. There was no significant weight loss until up to 300 °C. Thus, the TGA data demonstrated excellent thermal stability of the QDs-grafted COFs.

### 3.3. Kinetic Adsorption

The kinetic adsorption of 1 μmol L^−1^ of QCA was determined (Figure 3). After 60 min of shaking, 87.9% of QCA was adsorbed, and the equilibrium was achieved after 90 min. Therefore, QCA readily found the imprinted cavities on the MIPs based on QDs-grafted COFs surfaces, and the QDs-grafted COFs exhibited good mass transport.

### 3.4. Optosensing QCA Based on MIPs

QCA adsorption quenched the MIPs based on QDs-grafted COFs fluorescence in accordance with the Stern–Volmer equation:*F*_0_/*F*= 1 + *K*_SV_[*C*](1)
where *F*_0_ and *F* are the fluorescence intensity of MIPs based on QDs-grafted COFs in the absence and presence of QCA, respectively, *K*_SV_ is the Stern–Volmer constant, and [*C*] is the concentration of quencher QCA. 

The fluorescence was examined for various concentrations ranging over 0–0.1 mmol L^−1^. In Figure 4, the fluorescence intensity was reduced in the presence of QCA increasing concentrations. The fluorescence was quenched because of the H-bonding between the QDs-grafted COFs and QCA at the imprinted recognition sites. Analysis with the Stern–Volmer equation was performed in the linear range 1–50 μmol L^−1^, and the correlation was 0.9847. The detection limit was calculated as 0.85 μmol L^−1^, which was the concentration needed to quench the MIPs based on QDs-grafted COFs at a level three times the standard deviation of the blank signal to divide the slope of the standard curve. 

### 3.5. Interference Experiments for QCA Selectivity

The performance of MIPs based on QDs-grafted COFs was further evaluated by interference experiments, as shown in Figure 5. Several kinds of structural analogs, CBX, MEQ, MQCA, and QLA were used to demonstrate the selectivity of QDs-grafted COFs. Equilibrium binding experiments were performed for initial concentrations of 1–50 μmol L^−1^. The fluorescence intensity changes of MIPs based on QDs-grafted COFs with QCA were much greater than those without QCA. Because there were no tailor-made recognition sites on the MIPs based on QDs-grafted COFs formed QCA, less QCA was adsorbed. The coefficient ratio between MIPs based on QDs-grafted COFs with QCA and without QCA was 1.74. For the MIPs based on QDs-grafted COFs with QCA, it was mainly specific adsorption due to the selective recognition capability of tailor-made cavities, while it was mainly non-specific adsorption for the MIPs based on QDs-grafted COFs without QCA. Obviously, the MIPs based on QDs-grafted COFs had the highest binding affinity for QCA; however, there were different degrees of recognition for CBX, MEQ, MQCA, and QLA. The absorptions of CBX and QLA indicated no significant changes in fluorescence intensity, even though similar H-bonding can occur between the structural analogs and the APTES-modified QDs. The size and shape of MEQ and MQCA were more similar to that of QCA, and therefore it is possible that the molecule of MEQ and MQCA fitted neatly into the cavity of the MIPs and formed a hydrogen bond with the amino group of APTES on the MIPs, causing the MIPs to exhibit cross-binding ability. However, because of the lack of spatial complementarity, the degree of rebinding of the MIPs with MEQ and MQCA was smaller than that with QCA.

The mechanism may involve the interactions with the shape, size, and functionality of QCA. In addition, charge transfer from the luminescence QDs to QCA is caused by fluorescence quenching, as suggested by Tu et al. [25]. Furthermore, from the UV absorption spectrogram, the UV absorption band of QCA is near to the band-gap of the luminescence QDs, as revealed by UV-absorption spectra in Appendix A. Charges in the CdSe/ZnS QDs conduction bands can be transferred to the lowest unoccupied molecular orbital of QCA. Because there was no spectral overlap between QCA absorption and QD emission, energy transfer is unlikely.

### 3.6. Stability Analysis

To further analyze the analytical performance of the MIPs based on QDs-grafted COFs, stability was investigated (Appendix A). At room temperature, the MIPs based on QDs-grafted COFs were stable for nearly 60 days. This indicated that the MIPs based on QDs-grafted COFs possessed sufficient stability for QCA analysis. 

### 3.7. Detection of QCA in Meat and Feed Samples 

QCA detection in meat and feed samples was performed in a recovery experiment using 5, 10, and 20 μmol L^−1^ QCA. The recovery ranged from 93.8 to 101.2%, as shown in Table 1, indicating that the MIPs based on QDs-grafted COFs can be used to detect QCA accurately in complex samples.

## 4. Conclusions

This study reports the clear advantage of MIPs coated on QDs-based COFs for simple and rapid optosensing of QCA. The MIPs were prepared by a one-pot water-in-oil reverse microemulsion strategy. Here, QDs were used as an optical response element, and the COFs were used as an optosensing platform to combine the MIPs to provide high selectivity. The high sensitivity derived from the dual-signal amplification by high QD fluorescence quantum yields and the large COF-specific surface area. The results demonstrate that the MIPs based on QD-grafted COFs arrays detected QCA with high sensitivity and selectivity. The simple preparation, good specificity, low cost, good repeatability, and rapid detection make this approach attractive for hazard analysis.

## Figures and Tables

**Figure 1 polymers-11-00708-f001:**
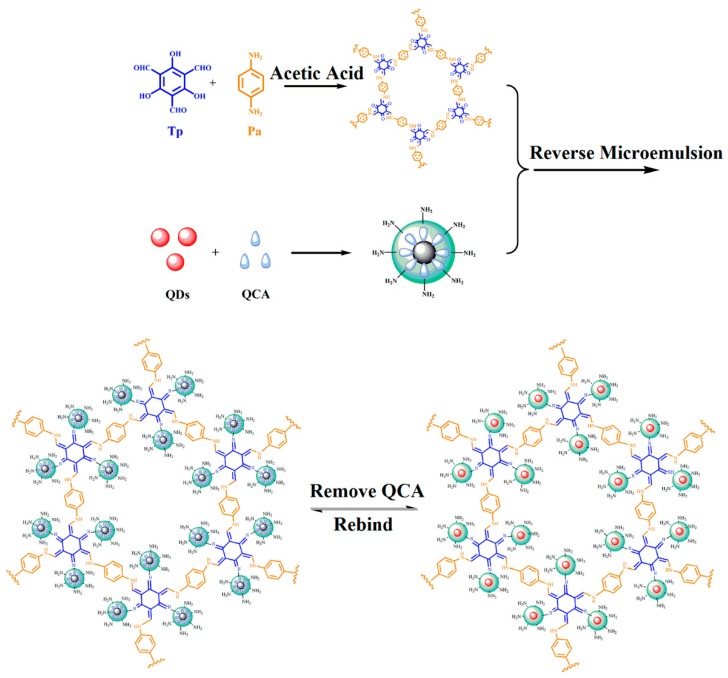
Scheme for the preparation of quantum dots (QDs)-grafted covalent organic frameworks (COFs).

**Figure 2 polymers-11-00708-f002:**
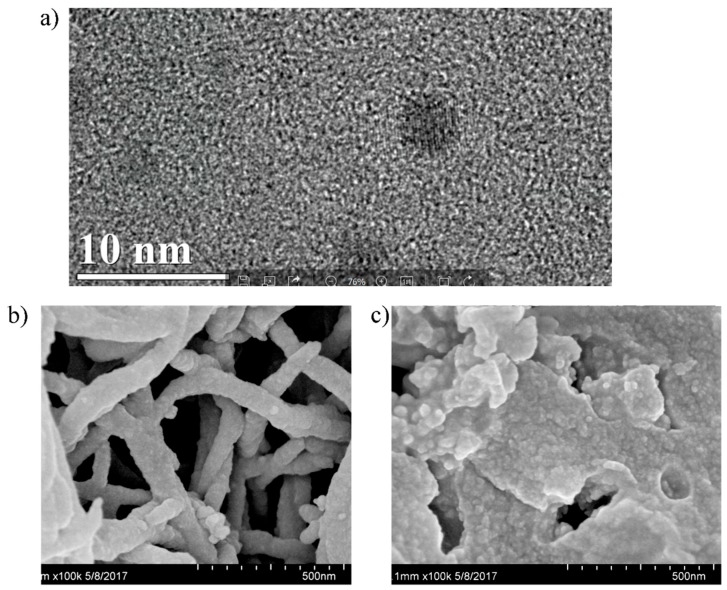
(**a**) TEM image of CdSe/ZnS QDs, (**b**) SEM image of COFs, (**c**) SEM image of QDs-grafted COFs.

**Figure 3 polymers-11-00708-f003:**
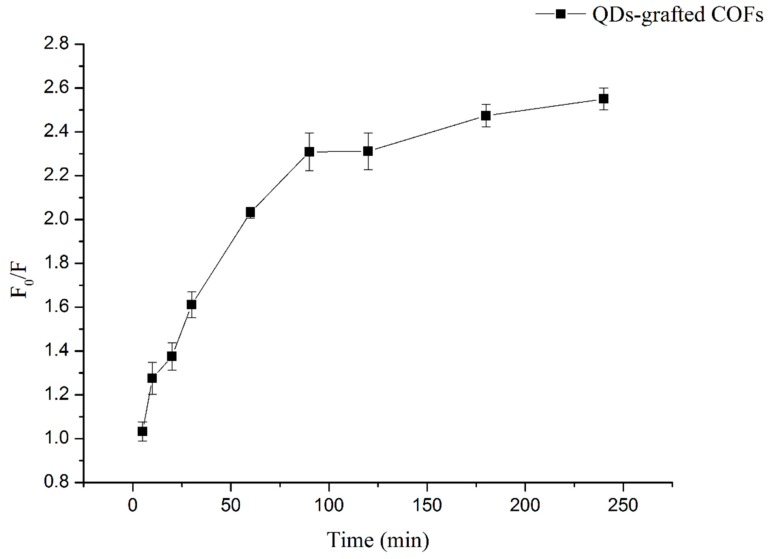
Kinetic adsorption of quinoxaline-2-carboxylicacid (QCA) (1 μmol L^−1^) onto QDs-grafted COFs.

**Figure 4 polymers-11-00708-f004:**
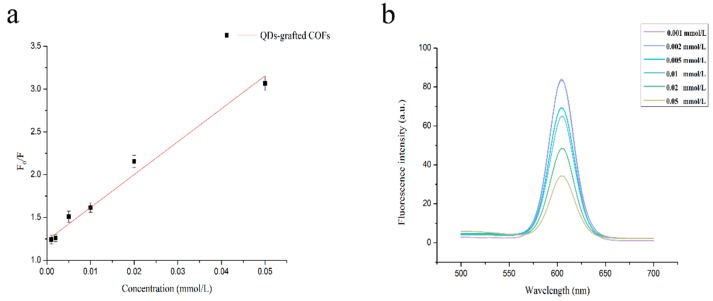
(**a**) Stern–Volmer plot of QCA concentration and (**b**) fluorescence intensity of QDs-grafted COFs.

**Figure 5 polymers-11-00708-f005:**
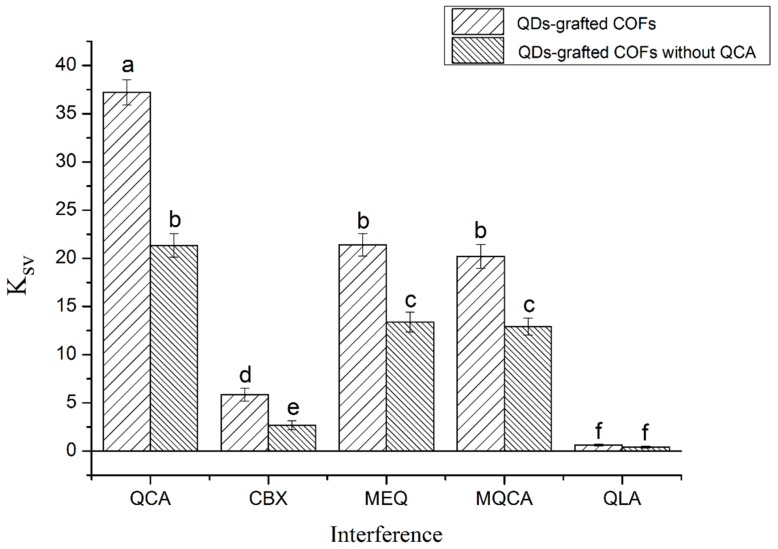
Results of the interferences of QCA and its structural analogs on QDs-grafted COFs.

**Table 1 polymers-11-00708-t001:** Spiked recovery results for QCA detection in real samples.

Samples	Added (μmol L^−1^)	Found (μmol L^−1^)	Recovery (%)	RSD (%)
Pork	0	-	-	-
	5	4.96	99.3	3.6
	10	9.83	98.3	4.3
	20	20.23	101.1	6.8
Beef	0	-	-	-
	5	4.83	96.6	3.3
	10	10.09	100.9	5.2
	20	19.77	98.8	4.1
Fish	0	-	-	-
	5	4.93	98.6	4.3
	10	9.97	99.7	5.8
	20	19.53	97.6	6.2
Pork feed	0	-	-	-
	5	4.94	98.8	6.7
	10	9.46	94.6	3.8
	20	19.37	96.9	5.4
Cattle feed	0	-	-	-
	5	4.72	94.4	4.7
	10	10.12	101.2	6.8
	20	18.73	93.8	3.9
Fish feed	0	-	-	-
	5	4.88	97.6	3.3
	10	9.98	99.8	5.1
	20	20.09	100.5	6.6

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
