# Peer review of "Luminescent Molecularly Imprinted Polymers Based on Covalent Organic Frameworks and Quantum Dots with Strong Optical Response to Quinoxaline-2-Carboxylicacid"

_polymers, 2019, doi:10.3390/polym11040708_

Round 1

Reviewer 1 Report

polymers-474756

The manuscript describes the development of MIPs for the detection of quinoxaline-2-carboxylicacid in meat and feed samples. A detection limit of 0.85 μmol L−1 and a linear concentration range between 1–50 μmol L−1 were reported. The MIPs based on quantum dots grafted covalent organic frameworks. In my opinion, this manuscript tackles a scientific relevant topic. The scientific and technical quality of this research seems to be good. The references seem to be appropriate and adequate. Therefore, I recommend the acceptance of this paper after a minor revision:

1) In the title is mentioned that quinoline-2-carboxylic acid is the target molecule. But in the abstract, the target molecule is charged to quinoxaline-2-carboxylicacid (QCA). Why? These are two different molecules.

2) Section 2.3.: Is the procedure adopted from other manuscript? If yes, please add the references.

3) Section 2.4.: Is the procedure adopted from other manuscript? If yes, please add the references.

4) In section 2.6: Please add that samples were analyzed before spiking for blank analysis.

5) The synthesis procedure explained in section 3.1. and figure1 needs to be clarified. First, figure 1 is too small which make it difficult to see the produced bonds and related atoms. Second, the stoichiometry in figure 1 is not correct. On the left there are 3 Tp and 2 Pa and on the right there are 6 Tp and 12 Pa (however in line 114 is more accurate with 0.3 mmol Tp and 0.45 mmol Pa).

6) The sentences in lines 155-156 and 160-162 must be rewritten in more detail which explain step-by-step occurring reactions and must be also fitted well to the figure1.

7) In figure 2: replace (c) with another picture having the same magnification like (b).

Author Response

Response to Reviewers

Reviewers' comments:

1. In the title is mentioned that quinoline-2-carboxylic acid is the target molecule. But in the abstract, the target molecule is charged to quinoxaline-2-carboxylicacid (QCA). Why? These are two different molecules.

Answer: I am very sorry about it. The target molecule is quinoxaline-2-carboxylicacid in the study. We have changed the title to “Luminescent molecularly imprinted polymers based on covalent organic frameworks and quantum dots with strong optical response on quinoxaline-2-carboxylicacid”. Thank you very much.

2. Section 2.3.: Is the procedure adopted from other manuscript? If yes, please add the references.

Answer: We have added the relevant reference in the Section 2.3. Thank you very much.

3. Section 2.4.: Is the procedure adopted from other manuscript? If yes, please add the references.

Answer: We have added the relevant reference in the Section 2.4. Thank you very much.

4. In section 2.6: Please add that samples were analyzed before spiking for blank analysis.

Answer: The samples analysis process was as follows. We have revised it in the manuscript. Thank you very much.

We chose the samples (5.00 ± 0.01 g) after it were homogenized by a high-speed food blender, to pack into a 100 mL of centrifuge tube. Then 5 mL of a solution containing 5% metaphosphoric acid (w/v) and 20% acetonitrile (v/v) was added. After the above mixture was mixed for 5 min, 10 mL of acetonitrile was added. Then it was centrifuged with 5000 ×g for 10 min, and the supernatant was collected and evaporated to near dryness at 40 °C. At last, the residue was re-dissolved within 1 mL of methanol and diluted to 100 mL of phosphate-buffered saline (pH 7.5).

5. The synthesis procedure explained in section 3.1. and figure1 needs to be clarified. First, figure 1 is too small which make it difficult to see the produced bonds and related atoms. Second, the stoichiometry in figure 1 is not correct. On the left there are 3 Tp and 2 Pa and on the right there are 6 Tp and 12 Pa (however in line 114 is more accurate with 0.3 mmol Tp and 0.45 mmol Pa).

Answer: We have enhanced the clarity of Figure 1. We used 0.3 mmol of Tp and 0.45 mmol of Pa to prepare TpPa COFs, and the molar ratio of Tp:Pa was 2:3, during the preparation. The TpPa COFs was used a reversible Schiff base reaction and an irreversible enol-to-keto tautomerization by aldehyde group (-CHO) of Tp and amino group (-NH3) of Pa. There were three -CHO and two –NH3 in Ta and Pa, respectively. But the expression is incorrect in Figure 1. I have revised it in the Fig.1. However, the structure of TpPa COFs was right, because of the structure of COFs was very large. The important part of COFs was showed in Fig.1, and we used wavy line to represent other structures. Thank you very much.

6. The sentences in lines 155-156 and 160-162 must be rewritten in more detail which explain step-by-step occurring reactions and must be also fitted well to the figure1.

Answer: We have revised the two sentences on line 185-190, and 195-199 of the revised manuscript. “Preparations of TpPa COFs were carried out by the Schiff-base reactions of Tp with Pa to the formation of a crystalline framework, in the presence of acetic acid using 1:1 mesitylene/dioxane as the solvent. The chemical stability of COFs was obtained by an irreversible enolto-keto tautomerization.” “APTES was used as the functional monomer to react with the target QCA using the H-bonding reactions. The APTES-modified QDs complex was further reacted with the TpPa COFs by the Schiff base reaction.” Thank you very much.

7. In figure 2: replace (c) with another picture having the same magnification like (b).

Answer: We have revised Fig.2c. Thank you very much.

Reviewer 2 Report

The submitted manuscript by Ying Zhang et al. describes the fabrication of quinoline-2-carboxylic acid (QCA) sensors based on superficially modified CdSe/ZnS quantum dots (QD) grafted to TpPa covalent organic framework. QDs are modified by QCA imprinted silica gel. The methodology has been previously described at T.Ni et al./SensorsandActuatorsB269(2018)340–345 by the authors and other co-workers and this previous publication should be mentioned and referred.

Concerning the results, I should mention that judging from low imprinting factor IF= 1,5 (which can be defined as ratio KsvMIP/KsvNIP, figure 5) the imprinting was not so successful. The selectivity for analogs MEQ and MQCA is also not so good (figure 5). And this should be discussed.   

The English language should be thoroughly revised. The conclusions presented in the manuscript should also be revised, otherwise, this is a set of not quite understandable phrases, e.g.

4. Conclusions

a) “Chemical and thermal stability of crystalline arrays of MIPs based on QD-grafted COFs were reported by a one-pot water-in-oil microemulsion method.”

What does it mean Chemical and thermal stability… were reported by one-pot water-in-oil microemulsion method????

b) “Here, QDs and COFs were used as an optical response element, and an optosensing platform, respectively, to combine the reverse microemulsion strategy to provide high selectivity.”

To combine microemulsion with what?

Other remarks:   

Page 4, lines 163-165

“Prior to the removal of the QCA, the fluorescence intensity of the QDs-grafted COFs was 37.4 %; after removal was 99.2 % (Fig. S1).” – it should be indicated that in comparison to non-imprinted polymer      

Page 5, lines 180-183

“Characteristic peaks at 1616 cm−1 are attributed to the decreased C=O stretching frequencies due to the covalent interactions of TpPa COFs and QDs. Therefore, the QDs were attached to the COF surface successfully.”

For correct judging of QDs successful grafting the FT-IR spectrum of COF should be presented for comparison

Page 6, figure 3 – the legend of the figure is not understandable

Page 7, figure 4 – in accordance with Stern–Volmer equation in the absence of QCA ([QCA]=0) the F0/F=1 , but not 1,25 as mistakenly indicated on the figure.  

Author Response

Response to Reviewers

Reviewers' comments:

1. The submitted manuscript by Ying Zhang et al. describes the fabrication of quinoline-2-carboxylic acid (QCA) sensors based on superficially modified CdSe/ZnS quantum dots (QD) grafted to TpPa covalent organic framework. QDs are modified by QCA imprinted silica gel. The methodology has been previously described at T.Ni et al./ Sensors and Actuators B 269 (2018)340–345 by the authors and other co-workers and this previous publication should be mentioned and referred.

Answer: We have added the relevant reference in the Section 2.4. Thank you very much.

2. Concerning the results, I should mention that judging from low imprinting factor IF= 1,5 (which can be defined as ratio KsvMIP/KsvNIP, figure 5) the imprinting was not so successful. The selectivity for analogs MEQ and MQCA is also not so good (figure 5). And this should be discussed.   

Answer: The coefficient ratio between MIPs based on QDs-grafted COFs with QCA and without QCA was 1.74. For the MIPs based on QDs-grafted COFs with QCA, it was mainly specific adsorption due to the selective recognition capability of tailor-made cavities, while it was mainly non-specific adsorption for the MIPs based on QDs-grafted COFs without QCA. Obviously, the MIPs based on QDs-grafted COFs had the highest binding affinity for QCA, but there are different degrees of recognition for CBX, MEQ, MQCA, and QLA. The absorptions of CBX, and QLA indicate no significant changes in fluorescence intensity, even though similar H-bonding can occur between the structural analogs and the APTES-modified QDs. The size and shape of MEQ and MQCA were more similar to that of QCA, and so it is possible that the molecule of MEQ and MQCA fitted neatly into the cavity of the MIPs and forms a hydrogen bond with the amino group in APTES on the MIPs, causing the MIPs to exhibit the cross-binding ability. However, because of the lack of spatial complementarity, the degree of rebinding the MIPs with MEQ and MQCA was smaller than that of QCA. We have discussed it on line 268-281 of the revised manuscript. Thank you very much.

3. The English language should be thoroughly revised. The conclusions presented in the manuscript should also be revised, otherwise, this is a set of not quite understandable phrases, e.g. 4. Conclusions

a) “Chemical and thermal stability of crystalline arrays of MIPs based on QD-grafted COFs were reported by a one-pot water-in-oil microemulsion method.”

What does it mean Chemical and thermal stability… were reported by one-pot water-in-oil microemulsion method????

Answer: We have revised the conclusions on line 302-305 of the manuscript. “This study reported clear advantage of MIPs coated on QDs-based COFs for simple and rapid optosensing QCA. The MIPs was prepared by a one-pot water-in-oil reverse microemulsion strategy.Thank you very much.

b) “Here, QDs and COFs were used as an optical response element, and an optosensing platform, respectively, to combine the reverse microemulsion strategy to provide high selectivity.”

To combine microemulsion with what?

Answer: We have revised the conclusions on line 302-305 of the manuscript. “Here, QDs were as an optical response element, and the COFs as an optosensing platform to combine the MIPs to provide high selectivity.” Thank you very much.

4. Other remarks:   

Page 4, lines 163-165

“Prior to the removal of the QCA, the fluorescence intensity of the QDs-grafted COFs was 37.4 %; after removal was 99.2 % (Fig. S1).” – it should be indicated that in comparison to non-imprinted polymer      

Answer: In the Fig.S1 (b) QDs-grafted COFs without QCA was non-imprinted polymer. Thank you very much.

Page 5, lines 180-183

“Characteristic peaks at 1616 cm−1 are attributed to the decreased C=O stretching frequencies due to the covalent interactions of TpPa COFs and QDs. Therefore, the QDs were attached to the COF surface successfully.”

For correct judging of QDs successful grafting the FT-IR spectrum of COF should be presented for comparison

Answer: We have added the FT-IR spectrum of COFs in the Supporting materials. The FT-IR spectra of COFs (Fig. S2a) and MIPs based on QDs-grafted COFs (Fig. S2b) is used to judge the occurrence of the polymerization reactions. It can be seen from Fig.S2a, the peaks at 1690 cm−1 and 1250 cm−1 were the stretching vibration of C=O and C-N, respectively, for the TpPa, and a strong peak at 1577 cm−1 attributed to the C=C stretch in the keto form. Characteristic peaks at 1616 cm−1 are attributed to the decreased C=O stretching frequencies due to the covalent interactions of TpPa COFs and QDs. Therefore, the QDs were attached to the COF surface successfully. We have added it on line 220-224 of the manuscript. Thank you very much.

Page 6, figure 3 – the legend of the figure is not understandable

Answer: We have changed the legend to “The kinetic adsorption of QCA (1 μmol L−1) onto QDs-grafted COFs”. Thank you very much.

Page 7, figure 4 – in accordance with Stern–Volmer equation in the absence of QCA ([QCA]=0) the F0/F=1 , but not 1,25 as mistakenly indicated on the figure.  

Answer: We have referred a lot of literatures about it, and verified the experimental results repeatedly. The Stern-Volmer equation did not pass through (0,1) points. We think it may be due to the experimental errors. In the experiment, the average value was obtained by repeating three times of the detection. Many papers have the similar results.

F0/F=0.0012x+1.0973 (A novel metronidazole fluorescent nanosensor based on graphene quantum dots embedded silica molecularly imprinted polymer. Mina Mehrzad-Samarin, Farnoush Faridbod, Amin Shiralizadeh Dezfuli, Mohammad Reza Ganjali. Biosensors and Bioelectronics, 2017, 92, 618–623.)

F0/F-1=0.01561[C]-0.13303 (Synthesis of surface molecular imprinting polymer on SiO2-coated CdTe quantum dots as sensor for selective detection of sulfadimidine. Zhiping Zhoua , Haiqin Yinga , Yanyan Liu, Wanzhen Xu, Yanfei Yang, Yu Luan, Yi Lu, Tianshu Liu, Shui Yu, Wenming Yang. Applied Surface Science, 2017, 404, 188-196.)

F0/F=0.0208[C]+1.0766 (Quantum dots coated with molecularly imprinted polymer as fluorescence probe for detection of cyphenothrin. Xiaohui Ren, Ligang Chen. Biosensors and Bioelectronics, 2015, 64, 182–188.)

F0/F=0.0219[C]+1.0769(A facile optosensing protocol based on molecularly imprinted polymer coated on CdTe quantum dots for highly sensitive and selective amoxicillin detection. Kochaporn Chullasat, Piyaluk Nurerk, Proespichaya Kanatharana, Frank Davis, Opas Bunkoed. Sensors and Actuators B, 2018, 254, 255–263.) Thank you very much.

Round 2

Reviewer 2 Report

After revision and corrections made I think that manuscript can be accepted for publication in Polymer